# Pressure stabilizes ferrous iron in bridgmanite under hydrous deep lower mantle conditions

Li Zhang [1] ✉, Yongjin Chen[1], Ziqiang Yang [1], Lu Liu [1], Yanping Yang[1], Philip Dalladay-Simpson [1], Junyue Wang[1] & Ho-kwang Mao [2]

Earth's lower mantle is a potential water reservoir. The physical and chemical properties of the region are in part controlled by the $Fe^{3+}/\Sigma Fe$ ratio and total iron content in bridgmanite. However, the water effect on the chemistry of bridgmanite remains unclear. We carry out laser-heated diamond anvil cell experiments under hydrous conditions and observe dominant $Fe^{2+}$ in bridgmanite $(Mg, Fe)SiO_3$ above 105 GPa under the normal geotherm conditions corresponding to depth > 2300 km, whereas $Fe^{3+}$-rich bridgmanite is obtained at lower pressures. We further observe FeO in coexistence with hydrous NiAs-type $SiO_2$ under similar conditions, indicating that the stability of ferrous iron is a combined result of $H_2O$ effect and high pressure. The stability of ferrous iron in bridgmanite under hydrous conditions would provide an explanation for the nature of the low-shear-velocity anomalies in the deep lower mantle. In addition, entrainment from a hydrous dense layer may influence mantle plume dynamics and contribute to variations in the redox conditions of the mantle.

There is a general consensus that the lower mantle is dominated by bridgmanite on the basis of high-pressure experimental data of density and sound velocities of the candidate minerals[1–3]. The models of the lower mantle are largely dependent on the chemical composition of bridgmanite throughout the entire lower mantle down to the top of D″ layer[4]. How the chemistry of bridgmanite especially the $Fe^{3+}/\Sigma Fe$ ratio and total iron content change with bulk composition under the lower mantle conditions remains controversial. Under pressure-temperature conditions of the topmost lower mantle, up to 16% $Fe^{3+}$ was obtained in bridgmanite $(Mg, Fe)SiO_3$ synthesized from $Fe^{2+}$-dominant material[5,6], whereas higher concentrations of $Fe^{3+}$ were observed in aluminous bridgmanite showing a nearly linear dependence of $Fe^{3+}/\Sigma Fe$ with $Al^{3+}$ content[6,7]. Furthermore, Fe metal was observed in coexistence with $Fe^{3+}$ as a result of the disproportionation of ferrous iron in bridgmanite[8].

Under pressure-temperature conditions of the deep lower mantle (>80 GPa), existing data of the $Fe^{3+}/\Sigma Fe$ ratio in $Al^{3+}$-bearing bridgmanite remain scattered ranging from 20 to 60% (Fig. S1) in part due to the differences in their starting materials[9–12]. The effects of pressure and $Al^{3+}$ content on the $Fe^{3+}/\Sigma Fe$ ratio of bridgmanite have not been fully clarified because of its complicating factors such as chemical composition, spin state of iron[10], difficult-to-achieve equilibrium of site distribution of iron[13], and possible iron oxidation induced by amorphization[7,14]. The ab initio calculations, however, suggested that the disproportionation reaction from ferrous iron to $Fe^{3+}$ plus iron metal is energetically favorable, in both Al-free and Al-rich compositions, at all lower mantle pressures[15]. Properties sensitive to the $Fe^{3+}/\Sigma Fe$ ratio of bridgmanite include element partitioning between lower mantle minerals[10,16], spin state of iron[17–19], and density and sound velocity profiles[3,20] under the lower mantle conditions. To clarify the effect of $Al^{3+}$ content on the $Fe^{3+}/\Sigma Fe$ in bridgmanite, we should further take into account the $H_2O$ effect as the presence of a hydrous phase could drastically reduce the $Al^{3+}$ content in bridgmanite with $Al^{3+}$ preferentially partitioning into the coexisting hydrous phase relative to

[1]Center for High Pressure Science and Technology Advanced Research, Shanghai, China. [2]Shanghai Key Laboratory MFree, Institute for Shanghai Advanced Research in Physical Sciences, Shanghai, China. ✉e-mail: zhangli@hpstar.ac.cn

bridgmanite[21–25]. To the best of our knowledge, the $H_2O$ effect on the chemistry of bridgmanite has never been reported. The effects of $Al^{3+}$ and $H_2O$ on the chemistry of bridgmanite under high-pressure-temperature conditions must be addressed in order to obtain an accurate model of the lower mantle and understand the origin of chemical heterogeneity in the deep lower mantle.

To understand the key factors controlling the iron oxidation state of bridgmanite, we designed experiments to separate the effects of $Al^{3+}$ and $H_2O$. First, we will determine the $Fe^{3+}/\Sigma Fe$ ratio of $Al^{3+}$-free bridgmanite as a function of pressure under hydrous conditions. Ortho-pyroxene (opx) $(Mg_{0.85}Fe_{0.15})SiO_3$ (Fs15) was sandwiched between hydrous silica gel as the starting material. Second, experiments on dry bridgmanite will be carried out for comparison. The previous study reported iron depletion in dry bridgmanite as a result of the disproportionation reaction[26]. Third, in order to evaluate the $Al^{3+}$ effect, we conducted one experiment on $Al^{3+}$-bearing bridgmanite in a hydrated basaltic composition to compare with the results of $Al^{3+}$-free bridgmanite under similar pressure-temperature conditions. Our experiments were performed in laser-heated diamond anvil cells over

the pressure and temperature range of 91-125 GPa and 1800-2400 K, close to mantle geotherm conditions[27].

## Results and discussions
### Experimental conditions

Experimental results and conditions are listed in Table 1. The phase assemblages and $Fe^{3+}/\Sigma Fe$ ratio of bridgmanite were obtained combining in situ X-ray diffraction (XRD) at high pressure with ex situ chemical analysis in a transmission electron microscope (TEM) on the samples recovered to ambient conditions (see "Methods"). A thin section suitable for TEM analysis was precisely lifted out from the heated center in each sample. The heated area can be clearly recognized under microscope in contrast to the surrounding transparent unreacted sample (Fig. 1). The homogeneity of color is an indication of no obvious variation in iron content across the heated spot. To further examine the effect of temperature gradient on the chemical composition, we managed to prepare a thin section across the temperature gradient of the heated spot along the radial direction and confirmed the homogeneity of the bulk composition (Fig. S2

**Table 1 | Experimental conditions and results**

| Run# | P&T | $P_{RT}$ | Sample/Medium | Phases by XRD | SAED | $Fe^{3+}/\Sigma Fe$ |
|---|---|---|---|---|---|---|
| 332-82 | 92 GPa&1850 K | 82 GPa | Fs15/h-silica | / | crystalline | 0.56(6) |
| 275-81 | 91 GPa&1950 K | 81 GPa | Fs15/h-silica | Brd+Nt+hcp-Fe | amorphous | $Fe^{3+}$ |
| 390-93 | 105 GPa&2250 K | 93 GPa | Fs15/h-silica | Brd + Nt | crystalline | 0.30(1) |
| 188-96 | 108 GPa&2250 K | 96 GPa | Fs15/h-silica | Brd + Nt | crystalline | 0.14(1) |
| 390-106 | 119 GPa&2400 K | 106 GPa | Fs15/h-silica | / | crystalline | 0.06(3) |
| 344-112 | 125 GPa&2400 K | 112 GPa | Fs15/h-silica | Brd + Nt | amorphous | $Fe^{2+}$ |
| 332-9582 | 107 GPa&2200 K | 95 GPa | Fs15/h-silica | Brd+Nt | / | / |
| decompress | 92 GPa&1900 K | 82 GPa | | Brd+Nt | amorphous | $Fe^{3+}$ |
| 344-99d | 112 GPa&2350 K | 99 GPa | Fs15/SiO₂ | Brd+H+Nt | / | / |
| 344-102d | 114 GPa&2200 K | 102 GPa | Fs15/SiO₂ | / | crystalline | / |
| 334-102d-2 | 115 GPa&2350 K | 102 GPa | Fs15/SiO₂ | / | crystalline | / |
| 344-108 m | 120 GPa&2200 K | 108 GPa | MAFSH/h-silica | / | unstable | 0.46-0.75 |
| 335-95 | 107 GPa&2300 K | 95 GPa | Fe(OH)₃/SiO₂ | FeO+Nt+py | / | $Fe^{2+}$ |
| 332-99 | 113 GPa&2500 K | 99 GPa | FSH/Ne | FeO+Nt+Ct | / | $Fe^{2+}$ |

Fs15: orthopyroxene $(Mg_{0.85}Fe_{0.15})SiO_3$; Brd: bridgmanite $(Mg, Fe)SiO_3$; Nt: NiAs-type $SiO_2$; H: H-phase; h-silica: hydrous silica gel containing ~2 wt.% $H_2O$; MAFSH: a gel sample with 24.9 mol% MgO-12.8 mol% $Al_2O_3$-7.5 mol% $Fe_2O_3$-54.8 mol% $SiO_2$ containing ~4 wt% $H_2O$; FSH: a hydrous gel sample with a molar ratio of $Fe_2O_3$: $SiO_2$ = 1:4 containing ~2 wt.% $H_2O$; Ct: $CaCl_2$-type $SiO_2$; py: pyrite-type $FeOOH_x$. The XRD measurements were performed on the samples at high pressure after temperature quench ($P_{RT}$). The numbers in parenthesis are one standard deviation.

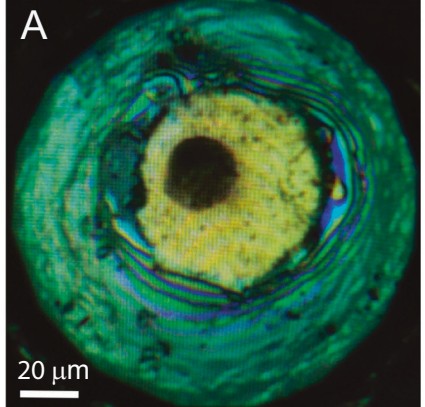
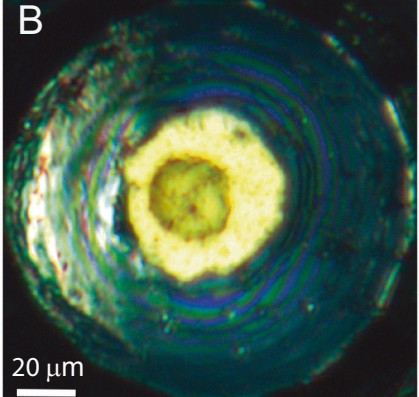

Run#344-112,125 GPa&2400 K          Run#275-81, 91 GPa&1950 K

**Fig. 1 | Representative microscopic images of the samples recovered to ambient conditions.** The heated area of the samples recovered from (**A**) 125 GPa and 2400 K (Run#344-112) versus (**B**) 91 GPa and 1950 K (Run#275-81).

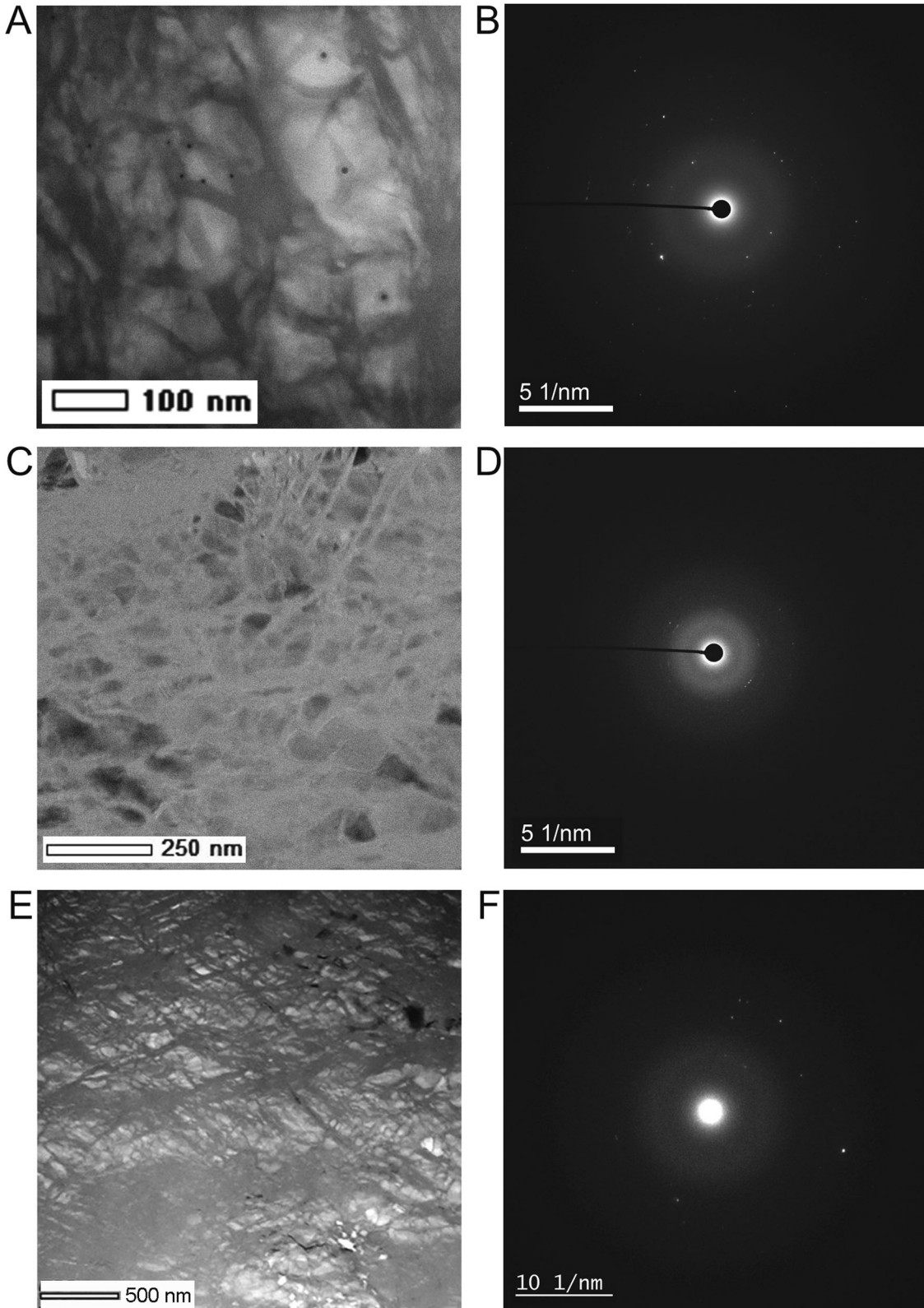

**Fig. 2 | TEM images and electron diffraction of the recovered crystalline bridgmanite. A, B** Run#390-93, showing a nearly pure bridgmanite phase with grain boundaries recovered from 105 GPa and 2250 K; (**C**) and (**D**) Run#188-96, recovered from 108 GPa and 2250 K; (**E**) and (**F**) Run#332-82, coexistence of bridgmanite and Fe metal recovered from 92 GPa and 1850 K. The SAED data confirmed crystallinity of bridgmanite in all the runs. The black dots in (**A**) are marks for damages induced by electron irradiation during the EELS measurement.

and Fig. S3). Element mapping and phase chemistry was obtained by energy dispersive spectroscopy (EDS). The bulk composition of the $5 \times 5$ µm² area in the heated center was obtained with $X_{Fe} = 0.15$, where $X_{Fe}$ is the iron content in atoms per two-cation formula unit, identical to the starting composition Fs15 (Fig. S3 and Table S1). Further, we carried out electron energy-loss spectroscopy (EELS) measurements at the Fe $L_{2,3}$-edges to distinguish between $Fe^{2+}$ and $Fe^{3+}$. Previous studies have demonstrated that EELS is an ideal tool for quantitative determination of $Fe^{3+}/\sum Fe$ ratio at the nanometer scale[7,14]. Specifically, to avoid possible iron oxidation induced by amorphization[7], we carefully examined crystallinity of bridgmanite before and after each EELS measurement using selected area electron diffraction (SAED) in a TEM. In this study, only those data from crystalline bridgmanite are used to examine the pressure effect on the $Fe^{3+}/\sum Fe$ ratio of bridgmanite.

## Pressure effect on the iron oxidation state of $Al^{3+}$-free bridgmanite under hydrous conditions

A thin layer of Fs15 opx was sandwiched between symmetric layers of hydrous silica gel containing ~2 wt.% $H_2O$. In the first run, the Fs15 sample was cold compressed to 93 GPa and then heated at 2250 K for 15 mins, corresponding to 105 GPa after accounting for the thermal pressure (Run#390-93, Table 1). The heating duration was 15 mins after the target temperature was reached in all the runs unless otherwise specified. The two-dimensional XRD scan on the sample after temperature quench showed the coexistence of bridgmanite and NiAs-type silica phases (Fig. S4). The sample was then recovered to ambient conditions and prepared for TEM analysis. The EDS mapping of the TEM section showed a homogeneous bridgmanite phase except for a few very small grains of iron metal (Fig. 2A). The chemical composition of bridgmanite was obtained based on multiple EDS analyses with $X_{Fe} = 0.14$ very close to the starting material Fs15 (Table S1), consistent with the observation of a nearly pure bridgmanite phase (Fig. 2A). The SAED data confirmed its crystallinity (Fig. 2B) and the EELS data (Fig. 3) revealed an $Fe^{3+}/\sum Fe$ ratio of 0.30(1) by a linear combination of Fe $L_3$ reference spectra of the ferrous and ferric iron standards (Fig. S5).

To evaluate the pressure effect on the $Fe^{3+}/\sum Fe$ ratio under hydrous conditions, we synthesized another two separate samples at 108 GPa and 2250 K (Run#188-96) and 119 GPa and 2400 K (Run#390-106), respectively (Table 1). The EELS data revealed the $Fe^{3+}/\sum Fe$ ratios of 0.14(1) and 0.06(3) for these two runs (Fig. 3), respectively, while the bridgmanite phase in both runs remained crystalline after the recovery (Fig. 2C–F). In the pressure range where bridgmanite is dominant in ferrous iron, we did not observe a temperature dependence of the iron content and $Fe^{3+}/\sum Fe$ ratio. In conclusion, the EELS data confirmed the stability of ferrous-iron-dominant bridgmanite above 105 GPa and 2250 K under hydrous conditions (Fig. 4).

In an experiment conducted at 91 GPa and 1950 K, the spotty diffraction pattern indicated the formation of a well-crystallized bridgmanite phase at high pressure (Fig. S2), but the TEM analysis on the recovered sample revealed an amorphous bridgmanite phase (Run#275-81, Fig. S6), implying that the sample lost its crystallinity during the recovery. The element mapping showed the coexistence of Fe-bearing bridgmanite, metallic particles and a silica phase (other than the pressure medium), consistent with in situ XRD observation at high pressure (Fig. S6). The EELS measurements revealed all $Fe^{3+}$ in the amorphous bridgmanite phase.

In order to recover a crystalline bridgmanite phase at this pressure, we synthesized another two samples at slightly lower temperatures. First, a reverse experiment was conducted (Run#332-9582). An $Fe^{2+}$-dominant bridgmanite phase was synthesized at 107 GPa and 2200 K based on the experimental conditions mentioned above. The sample was then decompressed to 82 GPa and heated again at 1900 K and 92 GPa for 20 mins. The recovered sample was an amorphous $Fe^{3+}$-

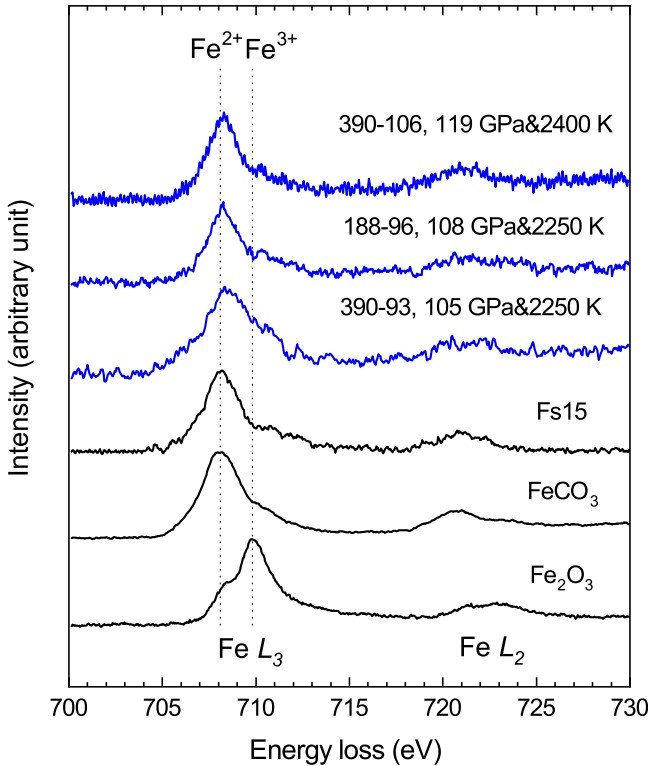

**Fig. 3 | EELS measurements showing ferrous-iron-dominant bridgmanite above 105 GPa and 2250 K under hydrous conditions.** The EELS data were recorded for the crystalline bridgmanite samples recovered from 105-119 GPa and 2250-2400 K, in comparison with the EELS data of the starting material Fs15, siderite $FeCO_3$ ($Fe^{2+}/\sum Fe = 100\%$) and hematite $Fe_2O_3$ ($Fe^{3+}/\sum Fe = 100\%$) measured under the same instrumental conditions. The maxima of $Fe^{2+}$ (708.1 eV) and $Fe^{3+}$ (709.8 eV) at the $L_3$-edge are indicated by the black dotted lines.

bridgmanite phase similar to the results of Run#275-81. We further conducted another experiment at 92 GPa and 1850 K (Run#332-82). Eventually, a crystalline bridgmanite phase was preserved after the recovery in coexistence with some iron metal (Fig. 2E, F). The EELS measurements on the crystalline bridgmanite phase revealed a mixture of $Fe^{2+}$ and $Fe^{3+}$ and small variations of the $Fe^{3+}/\sum Fe$ ratio was observed across the sample possibly due to the relatively low temperature for the synthesis (Fig. S7). We obtained an average $Fe^{3+}/\sum Fe$ ratio of 0.56(6) at 92 GPa and 1850 K, in comparison to dominant ferrous iron in bridgmanite above 105 GPa and 2250 K. The amorphization of bridgmanite in those runs at slightly higher temperatures might indicate the instability of the $Fe^{3+}$-dominant bridgmanite phase under ambient conditions. These results combined have showed a dramatic pressure effect on the $Fe^{3+}/\sum Fe$ ratio of bridgmanite under hydrous conditions with dominant ferrous iron in bridgmanite at depth greater than 2300 km (Fig. 4).

## Fe-bearing bridgmanite under dry versus hydrous conditions

To understand the role of $H_2O$ in controlling the iron oxidation state, we further compare the chemistry of bridgmanite between dry and hydrous conditions under similar pressure-temperature conditions. We conducted three separate sets of experiments in Fs15 sandwiched between dry silica layers at 112-115 GPa after accounting for the thermal pressure (Table 1). We conducted the first experiment at 114 GPa and 2200 K (Run#344-102d). The TEM images of the recovered sample showed the coexistence of Fe-depleted bridgmanite, a mixture of Fe-rich grain and a silica-rich amorphous phase (Fig. S3). The chemical analysis revealed Fe-depletion in bridgmanite with $X_{Fe} = 0.11$, while the bulk composition of $5 \times 5$ µm² area in the heated center is obtained with

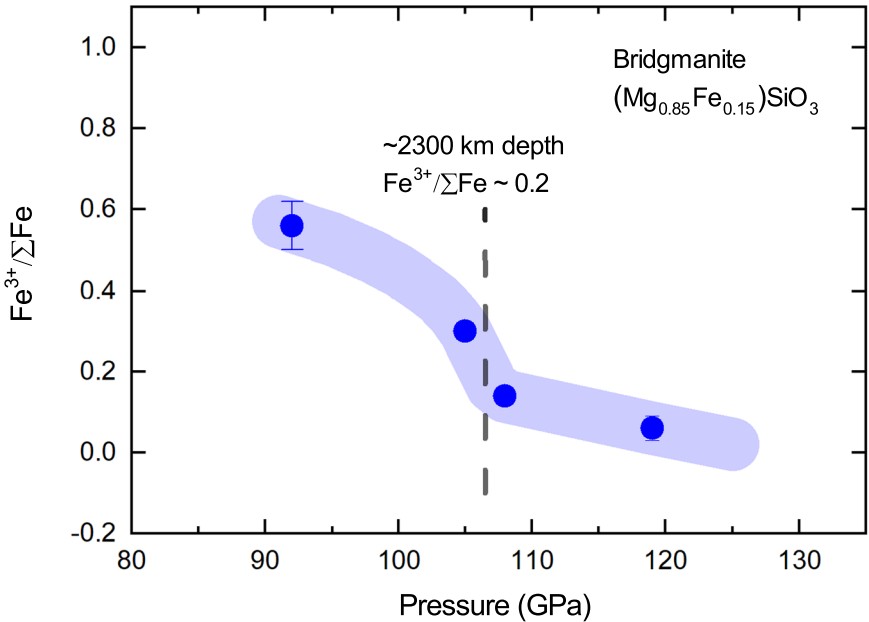

**Fig. 4 | Ferric iron concentration in bridgmanite under hydrous lower mantle conditions.** The blue solid circles represent the $Fe^{3+}/\Sigma Fe$ ratios measured in recovered crystalline bridgmanite samples over the pressure range of 90-130 GPa. The light blue curve is a guide to the eye for the $Fe^{3+}/\Sigma Fe$ ratio of bridgmanite. The dash line indicates the corresponding depth (~2300 km) where the $Fe^{3+}/\Sigma Fe$ ratio falls below 0.2.

$X_{Fe} = 0.15$, identical to the starting composition Fs15 (Table S1). To examine the temperature effect on Fe-depletion of dry bridgmanite, we conducted a separate experiment at a higher temperature of 2350 K and 115 GPa (Run#344-102d-2). We observed a greater Fe-depletion in bridgmanite with $X_{Fe} = 0.02$ at 2350 K compared to the run at 2200 K (Table S1). In agreement with the previous study[26], we confirmed Fe-depletion in bridgmanite as a result of the disproportionation reaction. The temperature effect on the disproportionation reaction and crystallization of the H-phase (Fig. S8) will require further investigation (see Supplementary Note 1 for details). The observation of Fe-depletion in bridgmanite under dry conditions, in contrast to the ferrous-iron-dominant bridgmanite without Fe loss under hydrous conditions, demonstrates that $H_2O$ stabilizes ferrous iron in bridgmanite under the deep lower mantle conditions at >2300 km depth.

To reveal how $H_2O$ stabilizes ferrous iron, we conducted experiments in the $Fe_2O_3$-$SiO_2$-$H_2O$ system under similar high-pressure-temperature conditions, and obtained FeO in coexistence with hydrous NiAs-type $SiO_2$ in the run products (Fig. 5), indicating that the stability of ferrous iron is a combined result of $H_2O$ effect and high pressure. At slightly lower pressures, a hexagonal hydrous phase $Fe_{12.76}O_{18}H_x$ was obtained in the $Fe_2O_3$-$H_2O$ and FeO-$H_2O$ systems, respectively[28]. The results further demonstrated that the iron valence state under $H_2O$-saturated deep lower mantle conditions is independent on the iron valence state in the starting materials.

## $Al^{3+}$ effect on the iron oxidation state of bridgmanite under hydrous conditions

The basaltic and pyrolitic compositions contain about 15 wt.% and 3-5 wt.% $Al_2O_3$, respectively[29]. To examine the $Al^{3+}$ effect on the iron oxidation state under hydrous conditions, we obtained aluminous bridgmanite phase in coexistence with an $Al^{3+}$-rich hydrous silica phase at 120 GPa and 2200 K using a hydrous gel starting material with all iron in $Fe^{3+}$. The MgO-$Al_2O_3$-$Fe_2O_3$-$SiO_2$ gel sample containing ~4 wt% $H_2O$ has a simplified basaltic composition and has been used in the previous studies[23,28]. In this sample, gradual amorphization of aluminous bridgmanite was observed, which led to an increase of the $Fe^{3+}/\Sigma Fe$ ratio from 0.46 to 0.75 between two consecutive EELS

measurements on one selected grain (Fig. S9). The results indicate that the $Fe^{3+}$ content of aluminous bridgmanite is coupled to its $Al^{3+}$ concentration under hydrous conditions. We obtained an atomic ratio Al/(Fe+Al) of ~0.40 in the recovered bridgmanite phase (Table S1), which is equal to the measured $Fe^{3+}/\Sigma Fe$ ratio within the uncertainties. About 60% of iron in aluminous bridgmanite is still in the $Fe^{2+}$ state despite all iron in $Fe^{3+}$ in the starting material. Furthermore, our observation showed that amorphization of aluminous bridgmanite could lead to an overestimation of its $Fe^{3+}/\Sigma Fe$ ratio. Future EELS measurements should be performed on crystalline aluminous bridgmanite to establish a relationship between $Fe^{3+}$ and $Al^{3+}$ content under hydrous lower mantle conditions.

Bridgmanite is nearly dry in coexistence with a hydrous phase[24]. Water can be stored in the high-pressure phases of silica in a basaltic composition[23,30–32]. Under the deep lower mantle conditions, the $Al^{3+}$-rich NiAs-type silica phase with an approximate formula $Si_{0.7}Al_{0.3}O_{1.85}H_x$[23] could contain up to 4.6 wt.% $H_2O$ via the $Si^{4+} = Al^{3+} + H^+$ charge-coupled substitution[30,33]. On the other hand, the solid solution of δ-phase[34] and phase H[35], AlOOH–$MgSiO_2(OH)_2$, was found stable in coexistence with bridgmanite in a pyrolitic lower mantle system[21,22,25]. In a system where the water content is lower than the level as simulated in our experiments, we would expect a decrease of water content in the hydrous phase or a smaller proportion of the hydrous phase. As $Al^{3+}$ preferentially partitions into the coexisting hydrous phase relative to bridgmanite[21–25], the $Fe^{3+}$ content in bridgmanite will be reduced accordingly due to the coupled substitution of $Fe^{3+}$ and $Al^{3+}$.

In summary, we investigated the combined effects of $H_2O$ and pressure on the chemistry of bridgmanite and obtained the following results: (1) ferric-iron-rich bridgmanite (Mg, Fe)$SiO_3$ was observed under hydrous conditions at depth <2000 km; (2) the presence of $H_2O$ in a coexisting hydrous phase stabilizes ferrous iron in bridgmanite at depth >2300 km, in contrast to Fe-depletion in dry bridgmanite (Mg, Fe)$SiO_3$ as a result of the disproportionation; and (3) our preliminary results at 120 GPa and 2200 K indicate that the $Fe^{3+}$ content is coupled to its $Al^{3+}$ concentration in bridgmanite under hydrous conditions. The experiments in a hydrated pyrolitic composition have not been conducted yet due to the technical challenges.

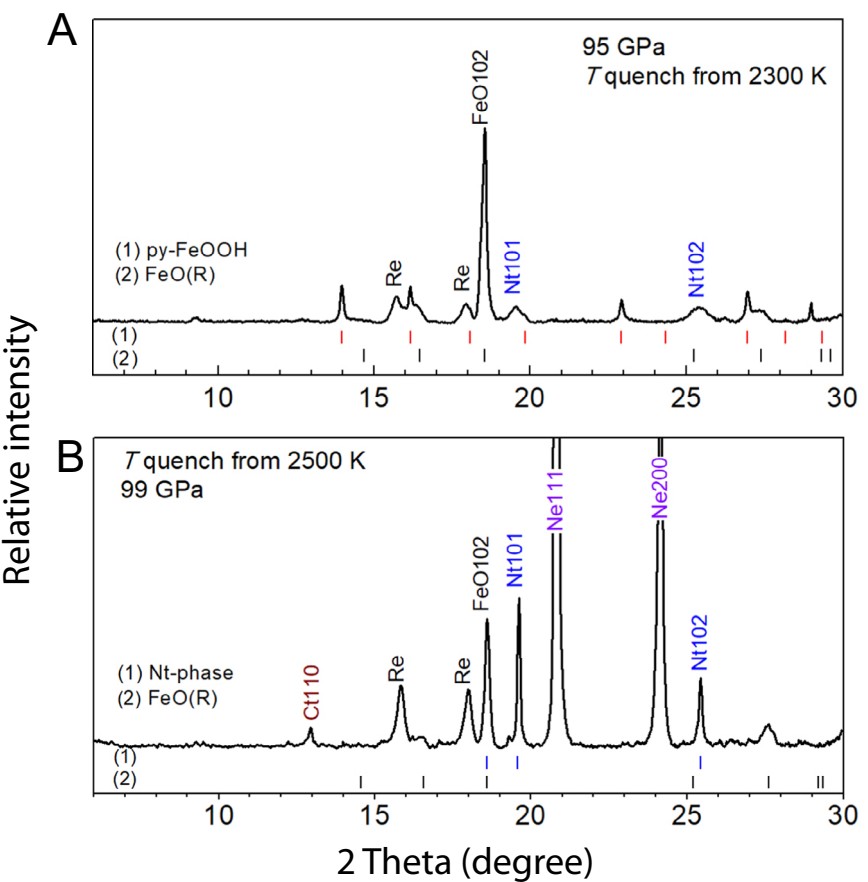

**Fig. 5 | Formation of the FeO phase in the $Fe_2O_3$-$SiO_2$-$H_2O$ system under high pressure-temperature conditions of the deep lower mantle. A** The starting material was $Fe(OH)_3$ sandwiched between dry $SiO_2$ layers (Run#335-95). The pyrite-structured $FeOOH_x$ (py-FeOOH) is in coexistence with rhombohedral(R) FeO when $H_2O$ is over-saturated in NiAs-type $SiO_2$ (Nt). **B** A hydrous gel sample with a molar ratio of $Fe_2O_3$: $SiO_2$ = 1:4 containing ~2 wt.% $H_2O$ was loaded in Ne medium (Run#332-99). The $CaCl_2$-type silica (Ct) is in coexistence with FeO(R) when $H_2O$ is unsaturated in the coexisting Nt phase. The X-ray wavelength was 0.6199 Å.

## Geophysical and geochemical implications

The deep lower mantle structure is dominated by two large low-shear-velocity provinces (LLSVPs) beneath the Pacific and Africa at depth greater than 2300 km[36–39]. The calculations of seismic velocities suggested that the LLSVPs with lowered shear-wave speeds and higher-than-average density can be well explained by iron enrichment in a bridgmanite dominant composition[40–42]. The stability of ferrous-iron-dominant bridgmanite under hydrous conditions, in contrast to the disproportionation and iron-depletion in dry bridgmanite[26], has important consequences for the deep lower mantle at depth >2300 km. Under deep lower mantle conditions above 60 GPa, the calculated shear-wave-velocity of iron silicate perovskite with composition 25 mol%$Fe_2O_3$-75mol%$FeSiO_3$ as a function of pressure[43] is nearly parallel to those of $MgSiO_3$ and $FeSiO_3$[44,45]. Considering iron-enrichment in bridgmanite from $X_{Fe}$ = 0.10 to 0.15, we obtain about 1.1% increase in density, 1.0% decrease in shear-wave velocity and 0.3% in bulk-sound velocity on the basis of the previous calculations[45]. The $H_2O$-induced iron-enrichment and stability of ferrous iron in bridgmanite is in general consistent with the character of the LLSVPs[46,47], providing an alternative to the basal magma ocean hypothesis[48]. To constrain geophysical and geochemical models of LLSVPs more quantitively, future research will be needed to examine Al and Fe partitioning between bridgmanite and coexisting phases in both hydrated basaltic and pyrolitic compositions, respectively. In particular, the occurrence of iron spin-pairing in ferropericlase[49–51] coupled with the $Fe^{3+}/\sum Fe$ ratio in bridgmanite could affect the Fe partitioning[10].

Importantly, the seismic imaging observations revealed a spatial connection between broad plume-like conduits rooted at the base of the mantle and major hotspots[52], and the correlation of hotspot locations within or at the borders of the LLSVPs further supports such a connection[53,54]. The presence of $H_2O$, even in small concentrations, strongly influences the rheological properties under the mantle conditions[55], although the rheological properties under the deep lower mantle conditions remain poorly understood. Whether the primordial noble gases and volatiles[56–58] are stored in LLSVPs is the subject of continuing debate[56,59]. However, a dense, low-viscosity layer at the base of the lower mantle may influence plume chemistry and dynamics and be critical in establishing the long-lived conduits in the lower mantle[60]. In addition, when the upwelling plumes contain such $H_2O$-bearing ferrous-iron-dominant material, disproportionation of ferrous iron would produce Fe metal plus ferric iron at a shallower depth, thus contributing to some variations in the redox conditions of the mantle.

## Methods

### Synchrotron X-ray diffraction

The experiments were conducted in laser-heated diamond anvil cells. Diamond anvils with flat culet diameters of 150 μm beveled at 10° up to 300 μm were used to generate pressure. Each sample was compressed to a target pressure and then heated using a double-sided heating system equipped with Ytterbium fiber lasers. The measured temperature uncertainties were within ±150 K[28]. Pressure was calibrated by the Raman shift of diamond anvil[61] at room temperature after temperature

quench. The thermal pressures can be estimated in this pressure range based on the equation $P_{th}$ (GPa) $= (T - 300) * 0.0062$[28]. The phase assemblages were characterized by XRD measurements conducted at 15U1 beamline of Shanghai Synchrotron Radiation Facility (SSRF) with an X-ray wavelength of 0.6199 Å or at the P02.2 beamline of PETRA III with an X-ray wavelength of 0.2900 Å.

## Transmission electron microscope (TEM) analysis

After a sample was recovered to ambient conditions, a cross-section was lifted from the center of heated area and thinned to 50–80 nm in thickness using a FEI Versa-3D focused ion beam (FIB). Elemental mapping and phase chemistry was obtained in a JEOL field emission TEM operating at 200 kV equipped with an EDS system. The EELS data were collected with an aperture of 5 mm, dispersion of 0.05 eV per channel, energy resolution of about 0.65 eV, dwell time from 0.01 to 0.1 s, and scan integration durations of 10–60 s. To enhance the signal-to-background ratio, the EELS data of Run#332-82 were collected with a dispersion of 0.15 eV per channel (Fig. S7). We determined the $Fe^{3+}/\sum Fe$ ratio in our recovered samples by a linear combination of Fe $L_3$ reference spectra of the ferrous ($FeCO_3$) and ferric iron ($Fe_2O_3$) standards from 703 to 717 eV following the method described by van Aken and Liebscher[14]. The numbers in parenthesis are one standard deviation based on multiple analyses. We obtained $Fe^{3+}/\sum Fe = 0.03(1)$ for our starting material Fs15 opx as a reference (Fig. S5).

## Reporting summary

Further information on research design is available in the Nature Portfolio Reporting Summary linked to this article.

## Data availability

All data generated this study are provided in the article or Supplementary Information. Source data for Figs. 3 and 5 are provided in the Supplementary Dataset. Source data are provided with this paper.

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

## Acknowledgements

We thank J. Li, E. Ohtani, and Q. Xia for discussions and comments. This work was supported by the National Natural Science Foundation of China (NSFC) (grant no. 42150103). H.-k.M. acknowledges financial support from Shanghai Key Laboratory Novel Extreme Condition Materials, China (no. 22dz2260800), Shanghai Science and Technology Committee, China (no. 22JC1410300), and National Science Foundation of China (grant no. U2230401). Portions of this work were performed at 15U1, Shanghai Synchrotron Radiation Facility (SSRF). Portions of this research were performed at the P02.2 beamline of PETRA III. We acknowledge DESY (Hamburg, Germany), a member of the Helmholtz Association HGF, for the provision of experimental facilities.

## Author contributions

L.Z. and H.-k.M. designed the project, L.Z., Z.Y., L.L. P.D.-S., and J.W. conducted the synchrotron experiments and data analysis, and Y.C., Y.Y., and L.Z. conducted the TEM analysis. All authors wrote the manuscript.

## Competing interests

The authors declare no competing interests.
