## [Peer Review File · Nature Communications]

Editorial Note: Figure on page 23 of this Peer Review File have been redacted as indicated to remove third-party material where no permission to publish could be obtained.

Reviewers' comments:

Reviewer #1 (Remarks to the Author):

The experiments conducted by the authors of this manuscript are very challenging and give new insight on possible chemical compositions of bridgmanite at very high-pressure and high-temperature. I have however quite a few concerns and I think that the authors did not report enough information for supporting their implications.

First of all, there is some misleading information in the introduction. The sentence at lines 26-28 "Previous models of the lower mantle have been built on the assumption that the ferric-to-ferrous iron ratio and chemical composition of bridgmanite remain unchanged throughout the entire lower mantle down to the top of the D" layer" is clearly wrong. The majority of models present in the literature (even the simplest ones as that reported in Huang et al. *Geochimica and Cosmochimica Acta* 303, 110-136, 2021) take into account Fe partitioning between ferropericlase and bridgmanite as well as several experimental observations which point out that the chemistry of bridgmanite is indeed changing with pressure and temperature. Moreover, in spite of correctly reporting that several studies have shown how the presence of Al in bridgmanite increases its Fe³⁺ content, the authors have chosen a Fe-bearing sample (i.e. devoid of Al), which clearly is not representative of a bulk lower mantle composition. No reason is given for such choice and surely the author cannot imply that there is enough water in the lower mantle to partition all Al present into hydrous phases! Moreover, the bridgmanite samples are present together with the high-pressure polymorph of stishovite, being such assemblage very different with what expected in a pyrolitic mantle. The implication given, therefore, are restricted to only special setting of the lower mantle and may be not representative at all.

Secondly, the authors claim that the synthesis have been performed in the presence of water and that this is likely present in the high-pressure polymorph of SiO₂. The fact that the heated spot is only a fraction of the pressure chamber suggests that the majority of water has likely migrated into the amorphous material surrounding the synthesis products. To suggest that hydrous conditions may be responsible together with pressure for the change in iron oxidation state is in my opinion largely overstretched.

Finally, no compositions of the bridgmanite samples synthesized at different conditions are reported. Although the authors claim that in one experimental run the Fe content of bridgmanite is similar to that of the starting material, it is difficult to assess such information without proper data (which have been anyway collected for all samples) and a proper assessment of uncertainties. Laser heating in a diamond anvil cell is known to produce severe Fe loss especially at such high temperatures and the authors should present any possible evidence of the contrary.

For all these reasons, I cannot recommend publication in Nature Communication without major revisions.

Reviewer #2 (Remarks to the Author):

This manuscript reported the ferric iron ratio ($\text{Fe}^{3+}/\text{total Fe}$) in bridgmanite in water-saturated conditions. As mentioned in the manuscript, bridgmanite is the main constituent mineral of the lower mantle and is the most abundant mineral on Earth. Thus the crystal chemistry of the bridgmanite is widely studied to understand the physical/chemical properties of the Earth's lower mantle.

In the manuscript, authors have shown that the ferric iron ratio of bridgmanite in wet conditions is above half at 90 GPa, and it decreases to 0.1 at a higher pressure than 100 GPa. It is concluded that the observed heterogeneity in the bottom of the lower mantle is attributed to the change in the ferric iron ratio of bridgmanite in wet conditions.

While the manuscript concluded that the change in the ferric iron ratio affects the physical and chemical properties of the mantle, the water-saturated Al-free bridgmanite is not likely the dominant material of the lower mantle.

Moreover, more importantly, the experimental results clearly contradict with authors' previous study showing the disproportionation of bridgmanite at similar conditions, and it is ignored in the current manuscript. The authors should fully explain why such a contradiction has occurred.

Overall I think this manuscript is not suitable for publication in Nature Communications.

Lower mantle

Any seismologic observation does not suggest that the lower mantle contains wt percent of water as simulated in the current experiments. The bridgmanite contains aluminum in pyrolite, and aluminum content largely controls the ferric-iron ratio. Since the current results are in water-saturated and Al-free bridgmanite, the implications are not suitable for the average mantle.

Contradiction in the studies

Authors previously reported bridgmanite is unstable at 95-100 GPa and 2200 - 2400K (Zhang et al., 2014). According to the results, bridgmanite with ~15% iron decomposes to iron-free bridgmanite and an iron-rich hexagonal phase (H-phase). Although the PT conditions and experimental apparatus are almost the same as in the current manuscript, the H-phase is not observed in the current results at all. Moreover, this contradiction is ignored in the manuscript. The absence of the H-phase in the current study suggests that the chemical equilibrium is not achieved in the current study (or in the previous study). It can be received with suspicions of duplicity. The authors should fully explain why such a contradiction has occurred. Otherwise, I am not convinced by any results shown by the authors.

Reviewer #3 (Remarks to the Author):

Experimental studies on the Fe^{2+} and Fe^{3+} ratios of bridgmanite under low mantle conditions were carried out in this study. The authors observed Fe^{2+} rich bridgmanite above ~105 GPa while Fe^{3+} rich

bridgmanite under lower pressure, along normal mantle geotherm temperature and in the presence of H₂O. They then discussed the possible chemical and physical implications of their observations. The experiments are hard to conduct, but the results are subject to the following issues:

a. Indeed, it has been documented that the lower mantle may be a water reservoir in the deep Earth, however, water is unlikely to be present as molecular H₂O as what was considered in this study. The mantle, including the lower mantle, is in general water unsaturated, and except for regional regions where H₂O-bearing fluids or melts may be present, the form of water that can be present in the deep mantle is mainly some species stored in silicate minerals, bound in their structures. The presence of H₂O molecules in the charges is inconsistent with "lower mantle" conditions. It is a question whether the results regarding the Fe²⁺/Fe³⁺ in bridgmanite can be applied to the lower mantle, and so are the implications.

b. Along subduction zone conditions where temperature is relatively cold, hydrous phases may be present, in coexisting with other normal mantle minerals such as bridgmanite. It has been shown that when coexisting with hydrous minerals (e.g., DHMS phases), the water content in normal mantle minerals could be extremely low (as indicated in Line 86 of the manuscript). However, the present work was carried out at geotherm conditions corresponding to the "normal" mantle, under those environments hydrous phases are unstable and should not form. Can explanations or discussions be provided for the presence of the hydrous phases in the experiments? Does this mean significant temperature uncertainty in the runs? More arguments should be provided for the reliability of the experiments and the data, since the results of the work are applied to the normal lower mantle. More importantly, it could be better if the water content in the synthesized samples is measured to make the results more convincing (though this might be difficult for the tiny samples in the charge...).

c. The discussions of the implications arising from the observations, Section 3, may be presented in more depth, in addition to the comments above for the presence of H₂O in the systems. The authors simply mentioned several aspects that have been proposed to be related to Fe²⁺ and/or Fe³⁺ in bridgmanite, but did not provide further quantitative analyses and in particular water if present in the lower mantle is unlikely to be H₂O. Given the molecular H₂O-free environments in the lower mantle, does the story still hold?

d. a minor issue: why a Mg_{0.85}Fe_{0.15}SiO₃ (Fs₁₅) opx was used, but not other compositions such as Fs₁₀ or others as often used for lower mantle bridgmanite? What is the particular reason for this choice? Will this influence the general results? These issues were not discussed.

We have provided the point-by-point responses including references following the reviewers'
comments and revised the main text and supplementary materials accordingly. Please note that our
responses are marked in black while the reviewers' comments are in blue. The new experimental
data in the revised version, **Table S1, Fig. S3 and Fig. S8-S9**, are shown as follows.

**Table S1. Comparison of chemical composition of the recovered bridgmanite phase under**
**hydrous versus dry deep lower mantle conditions.** The chemical composition of the
bridgmanite phase in each run were obtained using energy dispersive spectroscopy (EDS) in
different transmission electron microscopes (TEM), JEM-ARM200F or FEI Talos F200X, and
the chemical standards may slightly vary in different measurements. The numbers in parenthesis
are one standard deviation based on multiple analyses, where Mg# = 100 MgO/(MgO+FeO) in
mole. The bulk composition in the Run#344-102d was obtained with Mg# =85 in the 5×5 μm²
area of the heated center, an indication of no Fe loss compared to the starting material Fs15.

Run#		O (at.%)	Mg (at.%)	Si (at.%)	Fe (at.%)	Mg#
390-93	Bridgmanite/hydrous			105 GPa&2250 K		
	#1	60.30	17.64	19.53	2.53	
	#2	59.60	17.54	20.08	2.77	
	#3	60.27	17.29	20.05	2.39	
	#4	59.96	17.35	19.66	3.04	
	#5	60.66	17.02	19.35	2.97	
	Avg	60.16(0.14)	17.37(0.27)	19.73(0.20)	2.74(0.21)	86(1)
344-102d	Bridgmanite/dry			115 GPa&2350 K		
	#1	61.35	19.53	18.73	0.39	
	#2	61.57	19.53	18.64	0.26	
	#3	59.54	20.18	19.94	0.34	
	#4	61.36	19.93	18.46	0.26	
	Avg	60.96(0.82)	19.79(0.28)	18.94(0.58)	0.31(0.06)	98(0.3)
344-102d-2	Bridgmanite/dry			114 GPa&2200 K		
	#1	59.33	18.39	20.07	2.21	
	#2	59.18	18.26	20.35	2.21	
	#3	59.08	18.13	20.28	2.51	
	#4	58.66	18.92	20.38	2.05	
	Avg	59.06(0.25)	18.43(0.30)	20.27(0.12)	2.25(0.17)	89(1)
	Bulk	58.66	17.49	20.87	2.98	85
		O (at.%)	Mg (at.%)	Si (at.%)	Fe (at.%)	Al (at.%)
344-108m	Al ³⁺ -bearing bridgmanite/hydrous			120 GPa&2200 K		
	#1	59.48	12.3	18.66	6.18	3.38
	#2	58.32	12.4	18.87	6.12	4.29
	#3	59.94	11.03	17.87	6.70	4.46
	Avg	59.25(0.68)	11.91(0.62)	18.47(0.43)	6.33(0.26)	4.04(0.47)

**Fig. S3** Transmission electron microscope (TEM) images of the recovered sample showing the
coexistence of Fe-depleted bridgmanite, a mixture of Fe-rich grains (bright particles) and a silica-
rich amorphous phase (Run#344-102d, 114 GPa&2200 K). High-angle annular dark-field
(HAADF) scanning TEM images showing (A) a section over $\sim 15 \mu\text{m}$ length across more than half
of the heated spot along the radial direction and (B) a selected area in the heated center as indicated
by the rectangle box in (A). The chemical analysis listed in Table S1 indicates Fe-depletion in
bridgmanite with $\text{Mg}\# = 89$ ($\text{Mg}\# = 100 \text{MgO}/(\text{MgO}+\text{FeO})$ in mole) while the bulk composition
of the whole area in (B) remained identical to the starting material with $\text{Mg}\# = 85$.

**Fig. S8** X-ray diffraction pattern of Fs15 composition sandwiched between dry silica layers
collected at 99 GPa and after temperature quench from 2350 K (Run#344-99d). (A) A selected
area of the two-dimensional diffraction pattern and (B) the integrated diffraction pattern showing
weak peaks of the H-phase in coexistence with bridgmanite. Broad peaks of the niccolite-type (Nt)
structured silica indicate that the silica pressure medium was converted to the Nt-phase in a
metastable state at similar conditions for the observation of Nt-phase in dynamic compression
[Tracy *et al.*, 2020]. We obtained the unit-cell parameters for the bridgmanite (Brd) phase with a
$= 4.3568(7)$ Å, $b = 4.5913(15)$ Å, $c = 6.3373(15)$ Å and $V = 126.77(6)$ Å³ at 99 GPa. The X-ray
wavelength was 0.2900 Å.

**Fig. S9** TEM image and EELS data of the aluminous bridgmanite phase recovered from 120 GPa
and 2200 K under hydrous conditions. We used a gel starting material with 24.9 mol% MgO-12.8
38 mol% Al₂O₃-7.5 mol% Fe₂O₃-54.8 mol% SiO₂ containing ~4 wt% water (Run#344-108m). (A)
39 Coexistence of aluminous bridgmanite (Brd) with niccolite-type silica (Nt) phases. We conducted
two consecutive EELS measurements on one selected bridgmanite grain and collected SAED data
before and after each EELS measurement. We observed gradual amorphization of bridgmanite
during the measurement, which resulted in an increase of Fe³⁺/ΣFe from 0.46 (#1) to 0.75 (#2) as
shown in (B).

*Reviewers' comments:*

*Reviewer #1 (Remarks to the Author):*

*The experiments conducted by the authors of this manuscript are very challenging and give new*
*insight on possible chemical compositions of bridgmanite at very high-pressure and high-*
*temperature. I have however quite a few concerns and I think that the authors did not report*
*enough information for supporting their implications.*

Reply: Many thanks for recognizing the importance of our challenging experiments. We appreciate
the constructive comments and provide our point-by-point response as follows.

*First of all, there is some misleading information in the introduction. The sentence at lines 26-28*
*“Previous models of the lower mantle have been built on the assumption that the ferric-to-ferrous*
*iron ratio and chemical composition of bridgmanite remain unchanged throughout the entire*
*lower mantle down to the top of the D" layer” is clearly wrong. The majority of models present in*
*the literature (even the simplest ones as that reported in Huang et al. Geochimica and*
*Cosmochimica Acta 303, 110-136, 2021) take into account Fe partitioning between ferropericlasite*
*and bridgmanite as well as several experimental observations which point out that the chemistry*
*of bridgmanite is indeed changing with pressure and temperature.*

Reply: Following the suggestion, **L27-31**, the sentence is modified to “The models of the lower
mantle are largely dependent on the chemical composition of bridgmanite throughout the entire
lower mantle down to the top of D" layer[Huang et al., 2021]. How the chemistry of bridgmanite
especially the Fe³⁺/ ΣFe ratio and total Fe content change with bulk composition under the lower
mantle conditions remains controversial.”

*Moreover, in spite of correctly reporting that several studies have shown how the presence of Al*
*in bridgmanite increases its Fe³⁺ content, the authors have chosen a Fe-bearing sample (i.e.*
*devoid of Al), which clearly is not representative of a bulk lower mantle composition. No reason*
*is given for such choice and surely the author cannot imply that there is enough water in the lower*
*mantle to partition all Al present into hydrous phases!*

Reply: We conducted most of the experiments in Al-free samples in order to separate the effects
of Al³⁺ and H₂O on the iron oxidation state.

In the revised version, we added one experiment on Al³⁺-bearing bridgmanite using a hydrous gel
with all iron in Fe³⁺ as the starting material, to compare with the results of Al³⁺-free bridgmanite
under similar pressure-temperature conditions (**Fig. S9**). The details can be found in the new
section “**Al³⁺ effect on the iron oxidation state of bridgmanite under hydrous conditions**”
(**L175-208**). Under hydrous conditions of the deep lower mantle, the Fe³⁺ content in bridgmanite
is coupled to its Al³⁺ concentration while the rest of iron is still in the Fe²⁺ state.

**L179-185**, “To examine the Al³⁺ effect on the iron oxidation state under hydrous conditions, we
obtained aluminous bridgmanite phase in coexistence with an Al³⁺-rich hydrous silica phase at 120
85 GPa and 2200 K using a hydrous gel with all iron in Fe³⁺ as the starting material[Liu et al., 2022a;

*Liu et al., 2022b*]. In this sample, gradual amorphization of aluminous bridgmanite was observed
during the measurements, which led to an increase of the $\text{Fe}^{3+}/\Sigma\text{Fe}$ ratio from 0.46 to 0.75 between
two consecutive EELS measurements on one selected grain (Fig. S9).”

*Moreover, the bridgmanite samples are present together with the high-pressure polymorph of*
*stishovite, being such assemblage very different with what expected in a pyrolitic mantle. The*
*implication given, therefore, are restricted to only special setting of the lower mantle and may be*
*not representative at all.*

Reply: In our experiments, we loaded pure Fs15 sandwiched between silica layers. When a
pyrolitic starting material sandwiched between silica layers in the previous study, a coexistence of
bridgmanite and ferroperricite was obtained [*Sinmyo et al., 2011*]. In this study, we examined a
pure bridgmanite phase while the silica layers served as water supply.

In the revised version, we added one experiment on a hydrated basaltic composition to examine
the Al^{3+} effect (Run#344-108m) and included the new section “ **Al^{3+} effect on the iron oxidation**
**state of bridgmanite under hydrous conditions**”. We obtained the $\text{Fe}^{3+}/\Sigma\text{Fe}$ ratio of aluminous
bridgmanite nearly equal to the $\text{Al}/(\text{Fe}+\text{Al})$ atomic ratio at 120 GPa and 2200 K under hydrous
conditions with all iron in Fe^{3+} in the starting material.

The experiments in a hydrous pyrolitic system have not been conducted yet due to the technical
challenges.

*Secondly, the authors claim that the synthesis have been performed in the presence of water and*
*that this is likely present in the high-pressure polymorph of SiO_2 . The fact that the heated spot is*
*only a fraction of the pressure chamber suggests that the majority of water has likely migrated*
*into the amorphous material surrounding the synthesis products. To suggest that hydrous*
*conditions may be responsible together with pressure for the change in iron oxidation state is in*
*my opinion largely overstretched.*

Reply: Thanks for the constructive comments. To understand the role of water in controlling the
iron oxidation state, we further compared the chemistry of bridgmanite between dry and hydrous
conditions under similar pressure-temperature conditions. We added three sets of new experiments
on **dry bridgmanite** and a new section on “**Fe-bearing bridgmanite under dry versus hydrous**
**conditions**” (L143-174) in the revised version.

**L171-174**, “The observation of Fe-depletion in bridgmanite under dry conditions, in contrast to
the ferrous-iron-dominant bridgmanite without Fe loss under hydrous conditions, demonstrates
that H_2O stabilizes ferrous iron in bridgmanite under the deep lower mantle conditions at >2300
120 km depth.”

*Finally, no compositions of the bridgmanite samples synthesized at different conditions are*
*reported. Although the authors claim that in one experimental run the Fe content of bridgmanite*
*is similar to that of the starting material, it is difficult to assess such information without proper*
*data (which have been anyway collected for all samples) and a proper assessment of uncertainties.*

*Laser heating in a diamond anvil cell is known to produce severe Fe loss especially at such high*
*temperatures and the authors should present any possible evidence of the contrary.*

Reply: Thanks for the advice. We examined the chemical composition data for each experiment.
**Table S1** is added in the revised version, showing chemical composition of the recovered
bridgmanite phase under hydrous versus dry conditions. The bulk composition was obtained from
$5 \times 5 \mu\text{m}^2$ area in the heated center with Mg# =85, an indication of no Fe loss compared to the
starting material Fs15. (Table S1 and Fig. S3).

Fe loss was not observed in our experiments. **L79-86**: “To further examine the effect of a
temperature gradient on the chemical composition, we managed to prepare a thin section across
the temperature gradient along the radial direction of the heated spot and confirmed the
homogeneity of its chemical composition (Fig. S2 and Fig. S3). Element mapping and phase
chemistry was obtained by energy dispersive spectroscopy (EDS). The bulk composition of the
$5 \times 5 \mu\text{m}^2$ area in the heated center was obtained with Mg# =85 (Mg# = $100 \text{ MgO}/(\text{MgO} + \text{FeO})$ in
mole) identical to the starting composition Fs15 (Fig. S3 and Table S1).”

*For all these reasons, I cannot recommend publication in Nature Communication without major*
*revisions.*

*Reviewer #2 (Remarks to the Author):*
*This manuscript reported the ferric iron ratio ($\text{Fe}^{3+}/\text{total Fe}$) in bridgmanite in water-saturated*
*conditions. As mentioned in the manuscript, bridgmanite is the main constituent mineral of the*
*lower mantle and is the most abundant mineral on Earth. Thus the crystal chemistry of the*
*bridgmanite is widely studied to understand the physical/chemical properties of the Earth's*
*lower mantle.*

*In the manuscript, authors have shown that the ferric iron ratio of bridgmanite in wet conditions*
*is above half at 90 GPa, and it decreases to 0.1 at a higher pressure than 100 GPa. It is*
*concluded that the observed heterogeneity in the bottom of the lower mantle is attributed to the*
*change in the ferric iron ratio of bridgmanite in wet conditions.*

*While the manuscript concluded that the change in the ferric iron ratio affects the physical and*
*chemical properties of the mantle, the water-saturated Al-free bridgmanite is not likely the*
*dominant material of the lower mantle.*

*Moreover, more importantly, the experimental results clearly contradict with authors' previous*
*study showing the disproportionation of bridgmanite at similar conditions, and it is ignored in*
*the current manuscript. The authors should fully explain why such a contradiction has occurred.*
*Overall I think this manuscript is not suitable for publication in Nature Communications.*

Reply: We are glad that the Referee also noticed the contrast between the dry system (our
previous study) and the present wet system, that is exactly a major discovery of the current work.
The previous study [Zhang *et al.*, 2014] was conducted on dry bridgmanite while the current
study focuses on the effect of H₂O on the same sample. To reconfirm the contrast, we added

three new sets of experiments on dry bridgmanite (**Table S1 and Fig. S3**) and included a new
section “**Fe-bearing bridgmanite under dry versus hydrous conditions**” in the revised version
(**L143-174**). In agreement with the previous study [Zhang *et al.*, 2014], we confirmed Fe-
depletion in bridgmanite in coexistence with an Fe-rich phase under dry conditions.

**L171-174**, “The observation of Fe-depletion in bridgmanite under dry conditions, in contrast to
the ferrous-iron-dominant bridgmanite without Fe loss under hydrous conditions, demonstrates
that H₂O stabilizes ferrous iron in bridgmanite under the deep lower mantle conditions at >2300
173 km depth.”

The details and data are provided in the point-by-point responses as follows.

*Lower mantle*

*Any seismologic observation does not suggest that the lower mantle contains wt percent of water*
*as simulated in the current experiments. The bridgmanite contains aluminum in pyrolite, and*
*aluminum content largely controls the ferric-iron ratio. Since the current results are in water-*
*saturated and Al-free bridgmanite, the implications are not suitable for the average mantle.*

Reply: First, we conducted most of the experiments in Al-free samples in order to separate the
effects of Al³⁺ and H₂O on the iron oxidation state.

In the revised version, we added one experiment on Al³⁺-bearing bridgmanite using a hydrous gel
with all iron in Fe³⁺ as the starting material, to compare with the results of Al³⁺-free bridgmanite
under similar pressure-temperature conditions (**Fig. S9**). The details can be found in the new
section “**Al³⁺ effect on the iron oxidation state of bridgmanite under hydrous conditions**”
(**L175-208**). Under hydrous conditions of the deep lower mantle, the Fe³⁺ content in bridgmanite
is coupled to its Al³⁺ concentration while the rest of iron is still in the Fe²⁺ state.

Second, it is not practical to control the water content less than wt percent level in this type of
experiments. In the revised version, we added information about water storage (**L228-238**):

Most recently, water storage in the lower mantle has been extensively studied. Bridgmanite is
nearly dry in coexistence with a hydrous phase [Ishii *et al.*, 2022b]. Under the deep lower mantle
conditions, the Al³⁺-rich niccolite-type silica phase (Nt-phase) with an approximate formula
Si_{0.7}Al_{0.3}O_{1.85}H_x [Liu *et al.*, 2022] could contain up to 4.6 wt.% water via the Si⁴⁺ = Al³⁺ + H⁺
charge-coupled substitution [Ishii *et al.*, 2022a; Litasov *et al.*, 2007]. In a system where the water
content is lower than the level as simulated in our experiments, we would expect a decrease of
water content in the hydrous phase or a smaller proportion of the hydrous phase.

*Contradiction in the studies*

*Authors previously reported bridgmanite is unstable at 95-100 GPa and 2200 - 2400K (Zhang et*
*al., 2014). According to the results, bridgmanite with ~15% iron decomposes to iron-free*
*bridgmanite and an iron-rich hexagonal phase (H-phase). Although the PT conditions and*
*experimental apparatus are almost the same as in the current manuscript, the H-phase is not*
*observed in the current results at all. Moreover, this contradiction is ignored in the manuscript.*

*The absence of the H-phase in the current study suggests that the chemical equilibrium is not*
*achieved in the current study (or in the previous study). It can be received with suspicions of*
*duplicity. The authors should fully explain why such a contradiction has occurred. Otherwise, I*
*am not convinced by any results shown by the authors.*

Reply: Yes, the previous study [Zhang *et al.*, 2014] reported that “bridgmanite is unstable at 95-
100 GPa and 2200 - 2400K bridgmanite with ~15% iron decomposes to iron-free bridgmanite and
an iron-rich hexagonal phase (H-phase) in dry systems.”

In the section “**Fe-bearing bridgmanite under dry versus hydrous conditions**” in the revised
version (**L143-174**). We conducted three separate sets of experiments in Fs15 sandwiched between
dry silica layers (**Table 1**). We conducted the first experiment at 114 GPa and 2200 K (Run#344-
102d). The TEM images of the recovered sample showed the coexistence of Fe-depleted
bridgmanite, a mixture of Fe-rich grain and a silica-rich amorphous phase (Fig. S3).

We observed a temperature effect on the Fe-depletion in bridgmanite with Mg# = 89 at 2200 K
(Run#344-102d) and Mg# = 98 at 2350 K (Run#344-102d-2), respectively. Mg# = 100
MgO/(MgO + FeO) in mole

In agreement with the previous study[Zhang *et al.*, 2014], we confirmed Fe-depletion in
bridgmanite in coexistence with an Fe-rich phase in dry systems. **Table S1** is added in the revised
version, showing chemical composition of the recovered bridgmanite phase under hydrous versus
dry conditions.

In the revised version, Fe-depletion in dry bridgmanite is obtained in a solid pressure medium and
the temperature effect on the Fe-depletion is reported for the first time. However, crystallization
of the Fe-rich H-phase and its crystal chemistry is a very challenging project and we will discuss
the details in a separate study.

*Reviewer #3 (Remarks to the Author):*

*Experimental studies on the Fe²⁺ and Fe³⁺ ratios of bridgmanite under low mantle conditions*
*were carried out in this study. The authors observed Fe²⁺ rich bridgmanite above ~105 GPa*
*while Fe³⁺ rich bridgmanite under lower pressure, along normal mantle geotherm temperature*
*and in the presence of H₂O. They then discussed the possible chemical and physical implications*
*of their observations. The experiments are hard to conduct, but the results are subject to the*
*following issues:*

*a. Indeed, it has been documented that the lower mantle may be a water reservoir in the deep*
*Earth, however, water is unlikely to be present as molecular H₂O as what was considered in this*
*study. The mantle, including the lower mantle, is in general water unsaturated, and except for*
*regional regions where H₂O-bearing fluids or melts may be present, the form of water that can*
*be present in the deep mantle is mainly some species stored in silicate minerals, bound in their*
*structures. The presence of H₂O molecules in the charges is inconsistent with "lower mantle"*
*conditions. It is a question whether the results regarding the Fe²⁺/Fe³⁺ in bridgmanite can be*
*applied to the lower mantle, and so are the implications.*

Reply: Yes, water in the deep mantle is mainly stored in minerals structures. To avoid confusion,
we have modified “in the presence of H₂O” to “under hydrous conditions” or “in the presence of
a hydrous phase” in the revised version.

To clarify the effect of H₂O, we added a new section “**Fe-bearing bridgmanite under dry**
**versus hydrous conditions**” in the revised version (L143-174). In this section, we further
compared the chemistry of bridgmanite between dry and hydrous conditions under similar
pressure-temperature conditions.

*b. Along subduction zone conditions where temperature is relatively cold, hydrous phases may*
*be present, in coexisting with other normal mantle minerals such as bridgmanite. It has been*
*shown that when coexisting with hydrous minerals (e.g., DHMS phases), the water content in*
*normal mantle minerals could be extremely low (as indicated in Line 86 of the manuscript).*
*However, the present work was carried out at geotherm conditions corresponding to the*
*"normal" mantle, under those environments hydrous phases are unstable and should not form.*
*Can explanations or discussions be provided for the presence of the hydrous phases in the*
*experiments? Does this mean significant temperature uncertainty in the runs? More arguments*
*should be provided for the reliability of the experiments and the data, since the results of the*
*work are applied to the normal lower mantle. More importantly, it could be better if the water*
*content in the synthesized samples is measured to make the results more convincing (though this*
*might be difficult for the tiny samples in the charge...).*

Reply: First, it is not practical to control the water content less than wt percent level in this type
of experiments. Second, the water content in normal mantle minerals can be greatly enhanced by
both pressure and the Al content.

In the revised version, we added information about water storage (L228-238):

Most recently, water storage in the lower mantle has been extensively studied. Water can be stored
in the high-pressure phases of silica in a basaltic component [Ishii et al., 2022; Lin et al., 2022; Liu
et al., 2022b; Nisr et al., 2020]. Under the deep lower mantle conditions, the Al³⁺-rich niccolite-
type silica phase (Nt-phase) with an approximate formula Si_{10.7}Al_{0.3}O_{1.85}H_x [Liu et al., 2022b] could
contain up to 4.6 wt.% water via the Si⁴⁺ = Al³⁺ + H⁺ charge-coupled substitution [Ishii et al., 2022;
Litasov et al., 2007]. On the other hand, the solid solution of δ-phase [Sano et al., 2008] and phase
H [Nishi et al., 2014], AlOOH–MgSiO₂(OH)₂, was found stable in coexistence with bridgmanite in
a MgO-rich pyrolitic lower mantle system [Ohira et al., 2014; Walter et al., 2015; Yuan et al.,
2019]. In a system where the water content is lower than the level as simulated in our experiments,
we would expect a decrease of water content in the hydrous phase or a smaller proportion of the
hydrous phase.

*c. The discussions of the implications arising from the observations, Section 3, may be presented*
*in more depth, in addition to the comments above for the presence of H₂O in the systems. The*
*authors simply mentioned several aspects that have been proposed to be related to Fe²⁺ and/or*
*Fe³⁺ in bridgmanite, but did not provide further quantitative analyses and in particular water if*

*present in the lower mantle is unlikely to be H₂O. Given the molecular H₂O-free environments*
*in the lower mantle, does the story still hold?*

Reply: Thanks for the helpful advice. Based on two new sections “**Fe-bearing bridgmanite**
**under dry versus hydrous conditions**” (L143-174) and “**Al³⁺ effect on the iron oxidation**
**state of bridgmanite under hydrous conditions**” (L175-208) in the revised version, we
extended the implications accordingly.

*d. a minor issue: why a Mg_{0.85}Fe_{0.15}SiO₃ (Fs15) opx was used, but not other compositions*
*such as Fs10 or others as often used for lower mantle bridgmanite? What is the particular*
*reason for this choice? Will this influence the general results? These issues were not discussed.*

Reply: In EELS measurements, the intensities of Fe L_{2,3}-edges in the spectra are proportional to
the iron content in the samples. We chose Fs15 opx over Fs10 in our experiments in order to
obtain a good signal-to-background ratio for the Fe L_{2,3}-edges and achieved quantitative
determination of the Fe³⁺/ΣFe.

We added **Table S1**, showing chemical composition of the recovered bridgmanite phase under
hydrous versus dry deep lower mantle conditions.

The Fe content (16%) in the bridgmanite phase under hydrous conditions is close to the starting
material (15%) within the uncertainties (Table S1), consistent with the observation of a nearly pure
bridgmanite phase; we may infer that the Fe content in the bridgmanite phase would be close to
10% if Fs10 is used as the starting material.

**References:**

- Huang, R., T. Boffa Ballaran, C. A. McCammon, N. Miyajima, D. Dolejš, and D. J. Frost (2021),
The composition and redox state of bridgmanite in the lower mantle as a function of oxygen
fugacity, *Geochimica et Cosmochimica Acta*, 303, 110-136, doi:10.1016/j.gca.2021.02.036.
Ishii, T., G. Criniti, E. Ohtani, N. Purevjav, H. Fei, T. Katsura, and H.-k. Mao (2022),
Superhydrous aluminous silica phases as major water hosts in high-temperature lower mantle,
*Proceedings of the National Academy of Sciences*, 119(44), e2211243119,
doi:10.1073/pnas.2211243119.
Lin, Y., Q. Hu, M. J. Walter, J. Yang, Y. Meng, X. Feng, Y. Zhuang, R. E. Cohen, and H.-K.
Mao (2022), Hydrous SiO₂ in subducted oceanic crust and H₂O transport to the core-mantle
boundary, *Earth and Planetary Science Letters*, 594, 117708, doi:10.1016/j.epsl.2022.117708.
Litasov, K. D., H. Kagi, A. Shatskiy, E. Ohtani, D. L. Lakshmanov, J. D. Bass, and E. Ito (2007),
High hydrogen solubility in Al-rich stishovite and water transport in the lower mantle, *Earth and*
*Planetary Science Letters*, 262(3), 620-634, doi:10.1016/j.epsl.2007.08.015.
Liu, L., Z. Yang, H. Yuan, Y. Meng, N. Giordano, J. Sun, X. Du, P. Dalladay-Simpson, J. Wang,
and L. Zhang (2022a), Stability of a Mixed-Valence Hydrous Iron-Rich Oxide: Implications for
Water Storage and Dynamics in the Deep Lower Mantle, *Journal of Geophysical Research:*
*Solid Earth*, 127(5), e2022JB024288, doi:10.1029/2022JB024288.
Liu, L., H. Yuan, Y. Yao, Z. Yang, F. A. Gorelli, N. Giordano, L. He, E. Ohtani, and L. Zhang
(2022b), Formation of an Al-Rich Niccolite-Type Silica in Subducted Oceanic Crust:
Implications for Water Transport to the Deep Lower Mantle, *Geophysical Research Letters*,
49(15), e2021GL097178, doi:10.1029/2021GL097178.

Nishi, M., T. Irifune, J. Tsuchiya, Y. Tange, Y. Nishihara, K. Fujino, and Y. Higo (2014),
Stability of hydrous silicate at high pressures and water transport to the deep lower mantle,
*Nature Geoscience*, 7(3), 224-227, doi:10.1038/ngeo2074.
Nisr, C., H. Chen, K. Leinenweber, A. Chizmeshya, V. B. Prakapenka, C. Prescher, S. N.
Tkachev, Y. Meng, Z. Liu, and S.-H. Shim (2020), Large H₂O solubility in dense silica and its
implications for the interiors of water-rich planets, *Proceedings of the National Academy of*
*Sciences*, 117(18), 9747-9754, doi:10.1073/pnas.1917448117.
Ohira, I., E. Ohtani, T. Sakai, M. Miyahara, N. Hirao, Y. Ohishi, and M. Nishijima (2014),
Stability of a hydrous δ -phase, AlOOH–MgSiO₂(OH)₂, and a mechanism for water transport
into the base of lower mantle, *Earth and Planetary Science Letters*, 401, 12-17,
doi:10.1016/j.epsl.2014.05.059.
Sano, A., E. Ohtani, T. Kondo, N. Hirao, T. Sakai, N. Sata, Y. Ohishi, and T. Kikegawa (2008),
Aluminous hydrous mineral δ -AlOOH as a carrier of hydrogen into the core-mantle boundary,
*Geophysical Research Letters*, 35(3), doi:10.1029/2007GL031718.
Sinmyo, R., K. Hirose, S. Muto, Y. Ohishi, and A. Yasuhara (2011), The valence state and
partitioning of iron in the Earth's lowermost mantle, *Journal of Geophysical Research: Solid*
*Earth*, 116(B7), B07205, doi:10.1029/2010JB008179.
Tracy, S. J., S. J. Turneure, and T. S. Duffy (2020), Structural response of α -quartz under plate-
impact shock compression, *Science Advances*, 6(35), eabb3913, doi:10.1126/sciadv.abb3913.
Walter, M. J., A. R. Thomson, W. Wang, O. T. Lord, J. Ross, S. C. McMahon, M. A. Baron, E.
Melekhova, A. K. Kleppe, and S. C. Kohn (2015), The stability of hydrous silicates in Earth's
lower mantle: Experimental constraints from the systems MgO–SiO₂–H₂O and MgO–Al₂O₃–
SiO₂–H₂O, *Chemical Geology*, 418, 16-29, doi:10.1016/j.chemgeo.2015.05.001.
Yuan, H., L. Zhang, E. Ohtani, Y. Meng, E. Greenberg, and V. B. Prakapenka (2019), Stability
of Fe-bearing hydrous phases and element partitioning in the system MgO–Al₂O₃–Fe₂O₃–
SiO₂–H₂O in Earth's lowermost mantle, *Earth and Planetary Science Letters*, 524, 115714,
doi:10.1016/j.epsl.2019.115714.
Zhang, L., et al. (2014), Disproportionation of (Mg,Fe)SiO₃ perovskite in Earth's deep lower
mantle, *Science*, 344(6186), 877-882, doi:10.1126/science.1250274.

REVIEWER COMMENTS

Reviewer #1 (Remarks to the Author):

The authors have taken into account all my suggestions and have carefully revised the manuscript. In my opinion the manuscript has now the quality expected for an article published in Nature Communications

Reviewer #2 (Remarks to the Author):

The authors showed a new XRD pattern to confirm their previous work on discovering the H-phase (Figure S8). I was disappointed with the data. According to the earlier work, the peaks of the H-phase were sharp and intense (Zhang et al., 2014). However, the peaks of the H-phase in the new XRD pattern were obviously below the background level in a similar bulk composition (Figure S8). The data tells us that the experiments are not reproducible, unfortunately. It is natural to consider that the different results come from experiments with poor reproducibility rather than the H₂O content. Indeed, the H-phase is not reproduced by any other research group (Ismailova et al., 2016; Shim et al., 2017).

In addition, I can share the authors' opinion, saying that we need to separate the effects of aluminum and H₂O. At the same time, however, such a study will have a relatively niche impact. I think the study should be published in a more specific journal.

Reviewer #4 (Remarks to the Author):

This paper presents experimental evidence that ferrous iron (Fe²⁺) is the dominant iron species found in bridgmanite that has been synthesized above 100 GPa and at high temperatures under hydrous conditions. This assertion is novel because previous experiments have shown that Fe³⁺ is stable in bridgmanite due to disproportionation of Fe²⁺ and the generation of Fe metal. I have read through the previous round of reviewer comments. It seems that a major complaint raised by the previous reviewers was the lack of an Al-bearing composition, which is unrealistic in Earth's mantle. In response, the authors include 1 Al-bearing sample in this study and show that the Fe³⁺ content is dictated by the Al³⁺ content in a water-saturated environment. From what I understand, the punchline of the paper is that 1) in an aluminum-free, water-free environment, Fe partitions out of bridgmanite in favor of an Fe-rich H-phase, 2) in an aluminum-free, water saturated environment, Fe is stable in bridgmanite and the proportion of Fe³⁺ to total iron decreases as pressure is increased, and 3) when aluminum is present, the Fe³⁺ in bridgmanite is dictated by the Al content of the bridgmanite. Since the Al-bearing sample is so similar to the literature data they cite, it is hard for me to parse what is new and exciting here. Were the literature data also hydrous conditions? Did they have a mixture of Fe²⁺ and Fe³⁺? I think the paper would benefit a lot by the authors spelling out "this is what was done before, this is what is new and exciting". I do not recommend the manuscript for publication as is. I do not think more experiments need to be completed, but I do think the results need clarifying.

A couple of things I think could be addressed to improve the manuscript are:

- 1) What is the mechanism by which hydrous conditions would stabilize Fe²⁺ in bridgmanite? Particularly as bridgmanite is expected to be dry. Is it just that the Al is expected to be partitioned into a hydrous phase? If so, that contradicts the findings here since no Al-rich hydrous phase was discovered.
- 2) Is it possible to glean any information from the XRD patterns with regards to iron abundance within the bridgmanite or its oxidation state? Is there a volume dependence?
- 3) The possibility of a temperature dependence on total Fe and Fe³⁺/total Fe should be addressed. If they only expect a pressure dependence because of the atomic volume difference, that should be stated explicitly.
- 4) In the Al-bearing hydrous sample, the Fe started as entirely Fe³⁺ and some of it was reduced to Fe²⁺. Where did the electrons come from for that reduction? Something in the system must have been oxidized.
- 5) In the response to reviewers, the authors say that the Al³⁺-bearing sample had a basaltic starting composition, but that it was a hydrous gel. The mention of a basaltic composition is not noted anywhere in the manuscript (as far as I could tell); rather, they just say “an Al³⁺-bearing hydrous gel where Fe is entirely in the 3+ state”. This suggested to me that there was Fe in the gel, but not what its bulk composition was. How did they make such a composition?
- 6) The other reviewers pointed out that bridgmanite is not likely to be in equilibrium with SiO₂, but with (Mg,Fe)O. What would happen if ferropericlase was present in the starting material?
- 7) I agree with Reviewer 3 that the implications of this study are very vague. The authors make statements like “water in the lower mantle could result in lateral heterogeneity”. This is true of basically everything in the lower mantle. The authors could be a little more explicit about how hydrous conditions increase Fe in bridgmanite and seem to keep it reduced, and how that affects the shear wave speeds.

Review comments

Reviewer #1 (Remarks to the Author):

The authors have taken into account all my suggestions and have carefully revised the manuscript. In my opinion the manuscript has now the quality expected for an article published in Nature Communications

Reply: We appreciate the positive response from Referee#1. We are glad that the quality of the manuscript has been improved after the revision.

Reviewer #2 (Remarks to the Author):

The authors showed a new XRD pattern to confirm their previous work on discovering the H-phase (Figure S8). I was disappointed with the data. According to the earlier work, the peaks of the H-phase were sharp and intense (Zhang et al., 2014). However, the peaks of the H-phase in the new XRD pattern were obviously below the background level in a similar bulk composition (Figure S8). The data tells us that the experiments are not reproducible, unfortunately. It is natural to consider that the different results come from experiments with poor reproducibility rather than the H₂O content.

Indeed, the H-phase is not reproduced by any other research group (Ismailova et al., 2016; Shim et al., 2017).

Reply: The previous study [Zhang et al., 2014] was conducted on dry bridgmanite and reported that bridgmanite with ~15% iron decomposes to nearly Fe-free bridgmanite and an iron-rich hexagonal phase (H-phase).

In this study, we have confirmed the disproportionation of Fe-bearing bridgmanite and observed a greater Fe-depletion in bridgmanite with $X_{\text{Fe}}=0.02$ at 2350 K compared to the run with $X_{\text{Fe}}=0.11$ at 2200 K (Table S1), where X_{Fe} is the iron content in atoms per two-cation formula unit.

Chemical diffusion driven by a temperature gradient is commonly observed in laser-heated diamond anvil cell (LH-DAC) experiments [Andrault and Fiquet, 2001; Sinmyo and Hirose, 2010]. We managed to control chemical diffusion of (Mg,Fe)SiO₃ by improvement of the heating technique. In our previous round of response, we have provided detailed information to prove no Fe loss in the heated area of our experiments (**Table S1** and **Fig. S3**).

As Referee#2 pointed it out, research groups including [Ismailova et al., 2016; Shim et al., 2017] have conducted experiments on (Mg,Fe)SiO₃, but in the studies of [Ismailova et al., 2016] and [Shim et al., 2017], chemical analysis on the samples recovered from deep lower mantle conditions (>80 GPa) were not presented. **Without this critical information, it is impossible to evaluate chemical diffusion and iron distribution in their experiments.**

We further observed poor crystallization of both H-phase and dry bridgmanite in a solid pressure medium in this study whereas the peaks of the H-phase were sharp and intense in Ne pressure medium [Zhang et al., 2014]. Our progress on the H-phase will be discussed in a separate study. Since the crystallization of H-phase is not a focus of this study, we removed the related discussion of the H-phase from the manuscript and added **supplementary Note 1**:

“Supplementary Note 1. In an experiment conducted at 112 GPa and 2350 K (Run#344-99d), we performed XRD measurements at 99 GPa after temperature quench. Only weak diffraction peaks of the H-phase were observed while the texturing in the diffraction pattern of bridgmanite (**Fig. S8**) indicates a nanocrystalline sample with lattice preferred orientation possibly due to non-hydrostaticity in a solid pressure medium. No diffraction for any Fe metal or Fe oxides was observed. Weak diffraction of the H-phase, in contrast to a large proportion of Fe-rich component in coexistence with Fe-depleted bridgmanite in the recovered sample, implies poor crystallization of the H-phase in a solid pressure medium. In the previous study [Zhang *et al.*, 2014], a thin opx sample was loaded in Ne pressure medium and sharp diffraction peaks of the H-phase were observed.”

In addition, I can share the authors' opinion, saying that we need to separate the effects of aluminum and H₂O. At the same time, however, such a study will have a relatively niche impact. I think the study should be published in a more specific journal.

Reply: We appreciate that Referee#2 agreed on our revision regarding the Al³⁺ effect.

Reviewer #4 (Remarks to the Author):

This paper presents experimental evidence that ferrous iron (Fe²⁺) is the dominant iron species found in bridgmanite that has been synthesized above 100 GPa and at high temperatures under hydrous conditions. This assertion is novel because previous experiments have shown that Fe³⁺ is stable in bridgmanite due to disproportionation of Fe²⁺ and the generation of Fe metal. I have read through the previous round of reviewer comments. It seems that a major complaint raised by the previous reviewers was the lack of an Al-bearing composition, which is unrealistic in Earth's mantle. In response, the authors include 1 Al-bearing sample in this study and show that the Fe³⁺ content is dictated by the Al³⁺ content in a water-saturated environment.

Reply: We appreciate that Referee#4 recognized the novelty of our observation of dominant ferrous iron (Fe²⁺) above 100 GPa under hydrous conditions.

We obtained only one data in Al³⁺-bearing sample due to the technical difficulty (amorphization). For this reason, we deleted the sentence in the abstract, “Under hydrous conditions of the deep lower mantle, the Fe³⁺ content in bridgmanite is coupled to its Al³⁺ concentration while the rest of iron is still in the Fe²⁺ state.”

*From what I understand, the punchline of the paper is that 1) in an aluminum-free, water-free environment, Fe partitions out of bridgmanite in favor of an Fe-rich H-phase, 2) in an aluminum-free, water saturated environment, Fe is stable in bridgmanite and the proportion of Fe³⁺ to total iron decreases as pressure is increased, and 3) when aluminum is present, the Fe³⁺ in bridgmanite is dictated by the Al content of the bridgmanite. Since the Al-bearing sample is so similar to the literature data they cite, it is hard for me to parse what is new and exciting here. **Were the literature data also hydrous conditions? Did they have a mixture of Fe²⁺ and Fe³⁺?** I think the paper would benefit a lot by the authors spelling out “this is what was done before, this is what is*

new and exciting”. I do not recommend the manuscript for publication as is. I do not think more experiments need to be completed, but I do think the results need clarifying.

Reply: We agree that it is most important to clarify “*what was done before and what is new*”.

What is new: L55-56, To the best of our knowledge, the H₂O effect on the chemistry of bridgmanite has never been reported.

We have revised the main text to highlight “**what is new**” in our study, **L215-223**, “In summary, we investigated **the combined effects of H₂O and pressure on the chemistry of bridgmanite and obtained the following results**: (1) ferric-iron-rich bridgmanite (Mg,Fe)SiO₃ was observed under hydrous conditions at depth <2000 km; (2) the presence of H₂O in a coexisting hydrous phase stabilizes ferrous iron in bridgmanite at depth >2300 km, in contrast to Fe-depletion in dry bridgmanite (Mg,Fe)SiO₃ as a result of the disproportionation; and (3) our preliminary results at 120 GPa and 2200 K indicate that the Fe³⁺ content is coupled to its Al³⁺ concentration in bridgmanite under hydrous conditions. The experiments in a hydrated pyrolitic composition have not been conducted yet due to the technical challenges.”

What was done before: All the previous data of the Fe³⁺/ΣFe ratios was obtained in dry bridgmanite. (1) A nearly linear relationship between Fe³⁺/ΣFe with Al³⁺ content has been established in bridgmanite under conditions of **the topmost lower mantle** [Frost *et al.*, 2004; Lauterbach *et al.*, 2000; McCammon, 1997]. (2) At higher pressure of **deep lower mantle** (>80 GPa), the Fe³⁺/ΣFe ratios in Al³⁺-bearing bridgmanite are scattered ranging from 20 to 60% (**Fig. S1**).

A couple of things I think could be addressed to improve the manuscript are:

1) What is the mechanism by which hydrous conditions would stabilize Fe²⁺ in bridgmanite? Particularly as bridgmanite is expected to be dry. Is it just that the Al is expected to be partitioned into a hydrous phase? If so, that contradicts the findings here since no Al-rich hydrous phase was discovered.

Reply: Thanks for the constructive comments. **L172-179**, “To understand how H₂O stabilizes ferrous iron, we conducted experiments in the Fe₂O₃-SiO₂-H₂O system under similar high pressure-temperature conditions, and obtained FeO in coexistence with hydrous NiAs-type SiO₂ in the run products (Fig. S9), indicating that the stability of ferrous iron is a combined result of H₂O effect and high pressure. At slightly lower pressures, a hexagonal hydrous phase Fe_{12.76}O₁₈H_x was obtained in the Fe₂O₃-H₂O and FeO-H₂O systems, respectively [Liu *et al.*, 2022b]. The results further demonstrated that the iron valence state under H₂O-saturated deep lower mantle conditions is independent on the iron valence state in the starting materials.”

2) Is it possible to glean any information from the XRD patterns with regards to iron abundance within the bridgmanite or its oxidation state? Is there a volume dependence?

Reply: It is known that there is a linear relationship between the zero-pressure volume V₀ and iron abundance in bridgmanite, but we do not have zero-pressure volumes of the samples in our study.

We only decompressed the samples to ambient conditions right before the FIB/TEM preparation because the samples under hydrous conditions could be more likely oxidized in the air. Instead, we used chemical analysis to determine the Fe content.

3) The possibility of a temperature dependence on total Fe and Fe³⁺/total Fe should be addressed. If they only expect a pressure dependence because of the atomic volume difference, that should be stated explicitly.

Reply: **L120-121:** “In the pressure range where bridgmanite is dominant in ferrous iron, we did not observe a temperature dependence of the Fe content and Fe³⁺/ΣFe ratio.”

4) In the Al-bearing hydrous sample, the Fe started as entirely Fe³⁺ and some of it was reduced to Fe²⁺. Where did the electrons come from for that reduction? Something in the system must have been oxidized.

Reply: In our previous study [Liu et al., 2022b], Raman signal of O₂ was detected in the sample chamber where a nearly pure Fe_{12.76}O₁₈H_x phase was obtained at 78 GPa with Fe(OH)₃ as the starting material, implying that O₂ might be released when Fe³⁺ was reduced in a hydrous system. It was detected only once out of several attempts. We also used Raman to detect oxidized species in this study but nothing was found.

[Figure Redacted]

5) In the response to reviewers, the authors say that the Al³⁺-bearing sample had a basaltic starting composition, but that it was a hydrous gel. The mention of a basaltic composition is not noted anywhere in the manuscript (as far as I could tell); rather, they just say “an Al³⁺-bearing hydrous

gel where Fe is entirely in the 3+ state”. This suggested to me that there was Fe in the gel, but not what its bulk composition was. How did they make such a composition?

Reply: Silicate gels were prepared following the reported sol-gel procedure using reagent-grade $\text{Mg}(\text{NO}_3)_2 \cdot 6\text{H}_2\text{O}$, $\text{Al}(\text{NO}_3)_3 \cdot 9\text{H}_2\text{O}$, $\text{Fe}(\text{NO}_3)_3 \cdot 9\text{H}_2\text{O}$, and $(\text{C}_2\text{H}_5\text{O})_4\text{Si}$ [Hamilton and Henderson, 1968]. Fe_2O_3 in our gel sample is the decomposition product of $\text{Fe}(\text{NO}_3)_3 \cdot 9\text{H}_2\text{O}$.

The gel samples are widely used to ensure homogeneity of the starting material. For example, post-perovskite was first synthesized with MgSiO_3 gel as a starting material [Murakami et al., 2004].

L68-69, “a hydrous gel starting material” to “a hydrated basaltic composition”. After “using a hydrous gel starting material with all iron in Fe^{3+} ”, **L185-186**, we added “The $\text{MgO-Al}_2\text{O}_3\text{-Fe}_2\text{O}_3\text{-SiO}_2$ gel sample containing ~4 wt% H_2O has a simplified basaltic composition and has been used in the previous studies [Liu et al., 2022a; Liu et al., 2022b].”

And shown in the caption of Fig. S10, “We used a gel starting material with composition of 24.9 mol% MgO -12.8 mol% Al_2O_3 -7.5 mol% Fe_2O_3 -54.8 mol% SiO_2 containing ~4 wt% H_2O .”

6) The other reviewers pointed out that bridgmanite is not likely to be in equilibrium with SiO_2 , but with $(\text{Mg,Fe})\text{O}$. What would happen if ferropericlase was present in the starting material?

Reply: As we know ferropericlase contains only Fe^{2+} , we would expect to obtain partitioning of ferrous iron between bridgmanite and ferropericlase under hydrous conditions.

The community has a general concern about chemical diffusion driven by a temperature gradient [Andrault and Fiquet, 2001; Sinmyo and Hirose, 2010]. We have managed to control chemical diffusion of $(\text{Mg,Fe})\text{SiO}_3$ by improving the heating techniques and proved no Fe loss from the heating center (Table S1).

It is much more challenging to control heating in $(\text{Mg,Fe})\text{O}$ -bearing sample because Fe-Mg interdiffusion coefficients for perovskite are more than four orders of magnitude lower than those of ferropericlase at lower mantle conditions [Holzapfel et al., 2005]. We are working on a project to solve this problem.

7) I agree with Reviewer 3 that the implications of this study are very vague. The authors make statements like “water in the lower mantle could result in lateral heterogeneity”. This is true of basically everything in the lower mantle. The authors could be a little more explicit about how hydrous conditions increase Fe in bridgmanite and seem to keep it reduced, and how that affects the shear wave speeds.

Reply: We totally agree on the comments. We have revised the implications section to highlight **L225-259**: (1) The H_2O -induced iron-enrichment and stability of ferrous iron in bridgmanite would provide an explanation for the nature of the LLSVP anomalies at depth >2300 km and (2) Entrainment from a hydrous dense layer may influence mantle plume dynamics and contribute to variations in the redox conditions of the mantle.

References:

- Andrault, D., and G. Fiquet (2001), Synchrotron radiation and laser heating in a diamond anvil cell, *Review of Scientific Instruments*, 72(2), 1283-1288.
- Frost, D. J., C. Liebske, F. Langenhorst, C. A. McCammon, R. G. Trønnes, and D. C. Rubie

(2004), Experimental evidence for the existence of iron-rich metal in the Earth's lower mantle, *Nature*, 428(6981), 409-412.

Hamilton, D. L., and C. M. B. Henderson (1968), The preparation of silicate compositions by a gelling method, *Mineralogical Magazine and Journal of the Mineralogical Society*, 36(282), 832-838.

Holzappel, C., D. C. Rubie, D. J. Frost, and F. Langenhorst (2005), Fe-Mg Interdiffusion in (Mg,Fe)SiO₃ Perovskite and Lower Mantle Reequilibration, *Science*, 309(5741), 1707-1710.

Ismailova, L., et al. (2016), Stability of Fe,Al-bearing bridgmanite in the lower mantle and synthesis of pure Fe-bridgmanite, *Science Advances*, 2(7), e1600427.

Lauterbach, S., C. A. McCammon, P. van Aken, F. Langenhorst, and F. Seifert (2000), Mössbauer and ELNES spectroscopy of (Mg,Fe)(Si,Al)O₃ perovskite: a highly oxidised component of the lower mantle, *Contributions to Mineralogy and Petrology*, 138(1), 17-26.

Liu, L., H. Yuan, Y. Yao, Z. Yang, F. A. Gorelli, N. Giordano, L. He, E. Ohtani, and L. Zhang (2022a), Formation of an Al-Rich Niccolite-Type Silica in Subducted Oceanic Crust: Implications for Water Transport to the Deep Lower Mantle, *Geophysical Research Letters*, 49(15), e2021GL097178.

Liu, L., Z. Yang, H. Yuan, Y. Meng, N. Giordano, J. Sun, X. Du, P. Dalladay-Simpson, J. Wang, and L. Zhang (2022b), Stability of a Mixed-Valence Hydrous Iron-Rich Oxide: Implications for Water Storage and Dynamics in the Deep Lower Mantle, *Journal of Geophysical Research: Solid Earth*, 127(5), e2022JB024288.

McCammon, C. (1997), Perovskite as a possible sink for ferric iron in the lower mantle, *Nature*, 387(6634), 694-696.

Murakami, M., K. Hirose, K. Kawamura, N. Sata, and Y. Ohishi (2004), Post-Perovskite Phase Transition in MgSiO₃, *Science*, 304(5672), 855-858.

Shim, S.-H., B. Grocholski, Y. Ye, E. E. Alp, S. Xu, D. Morgan, Y. Meng, and V. B. Prakapenka (2017), Stability of ferrous-iron-rich bridgmanite under reducing midmantle conditions, *Proceedings of the National Academy of Sciences*, 114(25), 6468-6473.

Sinmyo, R., and K. Hirose (2010), The Soret diffusion in laser-heated diamond-anvil cell, *Physics of the Earth and Planetary Interiors*, 180(3), 172-178.

Zhang, L., et al. (2014), Disproportionation of (Mg,Fe)SiO₃ perovskite in Earth's deep lower mantle, *Science*, 344(6186), 877-882.

REVIEWERS' COMMENTS

Reviewer #4 (Remarks to the Author):

The authors have addressed my concerns and, in my opinion, the manuscript is ready for publication. I particularly like the new discussion section. It is much more specific.

There is one minor grammatical error:

Line 209-210: "preferentially partitions" instead of "preferentially partitioning"